# Stochastic Finite Element Analysis Framework for Modelling Mechanical Properties of Particulate Modified Polymer Composites

**DOI:** 10.3390/ma12172777

**Published:** 2019-08-29

**Authors:** Hamidreza Ahmadi Moghaddam, Pierre Mertiny

**Affiliations:** Department of Mechanical Engineering, University of Alberta, Edmonton, AB T6G 1H9, Canada

**Keywords:** stochastic finite element analysis, filler modified polymer composites, modulus of elasticity, Poisson’s ratio, thermal expansion coefficient

## Abstract

Polymers have become indispensable in many engineering applications because of their attractive properties, including low volumetric mass density and excellent resistance to corrosion. However, polymers typically lack in mechanical, thermal, and electrical properties that may be required for certain engineering applications. Therefore, researchers have been seeking to improve properties by modifying polymers with particulate fillers. In the research presented herein, a numerical modeling framework was employed that is capable of predicting the properties of binary or higher order composites with randomly distributed fillers in a polymer matrix. Specifically, mechanical properties, i.e., elastic modulus, Poisson’s ratio, and thermal expansion coefficient, were herein explored for the case of size-distributed spherical filler particles. The modeling framework, employing stochastic finite element analysis, reduces efforts for assessing material properties compared to experimental work, while increasing the level of accuracy compared to other available approaches, such as analytical methods. Results from the modeling framework are presented and contrasted with findings from experimental works available in the technical literature. Numerical predictions agree well with the non-linear trends observed in the experiments, i.e., elastic modulus predictions are within the experimental data scatter, while numerical data deviate from experimental Poisson’s ratio data for filler volume fractions greater than 0.15. The latter may be the result of morphology changes in specimens at higher filler volume fractions that do not comply with modelling assumptions.

## 1. Introduction

Having low volumetric mass density and high environmental degradation resistance (e.g., corrosion), while being low-cost, polymers are attractive for a variety of engineering applications [1,2,3]. Yet, limited mechanical, thermal, and electrical properties often limit the usability of polymers. This problem has motivated material designers to enhance polymer properties by adding appropriate types of fillers [4]. A variety of filler types and morphologies, in addition to the chosen filler volume fractions (FVF), are available for modifying polymers. The multitude of micro and nano-scale particles, and combinations thereof, forming not only single but binary, ternary, and even higher order modified polymers, create a need for modeling and experimental processes to guide the design of filler-modified polymers with well dispersed and distributed particles [5,6]. As such, the development of filler modified polymer composites is a costly and time-consuming process that is challenging to perform [7] by material designers.

Various studies have been reported for predicting the properties of advanced materials [8,9,10,11]. Eshelby pioneered the characterization of particulate composites by performing stress field analysis with an ellipsoidal inclusion [12]. The Halpin-Tsai empirical model, as shown in Equations (1) and (2), was proposed for predicting the modulus of elasticity for a variety of particle geometries [13].
(1)$$E_{c}=E_{m}\frac{1+\zeta \eta \phi }{1-\eta \phi }$$
(2)$$\eta =\left(\frac{E_{f}}{E_{m}}-1\right){\left(\frac{E_{f}}{E_{m}}+\zeta \right)}^{-1}$$
where *E*_f_ and *E*_m_ are the modulus of elasticity of filler and matrix respectively, *ϕ* is the volume fraction of filler, and *ζ* and *E*_c_ are correspondingly a shape factor and the effective modulus of elasticity.

The approximation by Mori and Tanaka [14] is an extension of the Eshelby solution. Other analytical methods include modelling approaches by Benveniste [15], Christensen and Lo [16], and Torquato [17].

Some analytical approaches provide estimates for the upper and lower bounds of the modulus of elasticity with respect to the filler volume fraction. The Voight [18] and Reuss [19] models, as indicated by Equations (3) and (4), are the most basic upper and lower bounding techniques, respectively. The variational approach developed by Hashin and Shtrikman [20] is another popular analytical technique for predicting the bounds for the effective modulus of elasticity.
(3)$$E_{U}=\phi E_{f}+\left(1-\phi \right)E_{m}$$
(4)$$\frac{1}{E_{L}}=\frac{\phi }{E_{f}}+\frac{\left(1-\phi \right)}{E_{m}}$$
where *E*_U_ and *E*_L_ are the upper and lower bound effective modulus of elasticity, respectively.

In addition to randomly oriented particles, modified polymers with aligned particles have gained interest among material designers for improving mechanical and thermal properties [21]. High strength filler modified polymers [22,23,24] and sensors [25,26] are example applications that require filler alignment to create explicit anisotropic mechanical properties. Carbon nanotubes, for example, are a popular filler type being considered for creating anisotropic materials for a variety of applications [27,28].

Analytical approaches are widely recognized as expedient for predicting mechanical properties of filler modified polymers. However, these methods often lack in accuracy, especially for high FVF. Experimental studies, on the other hand, provide direct information on material properties. However, being time-consuming and costly, experiments by themselves are an unsatisfactory solution for researchers designing advance materials [29]. In light of the aforementioned complexities associated with filler modified polymers, and a rapidly growing number of opportunities for these composites in the industry, it is imperative to identify alternative approaches that allow for the efficient and effective prediction of material properties [30,31,32].

Stochastic methods are prominently used in reliability analysis, which requires analyzing multiple input variables and predicting outcomes with a suitable level of accuracy. Stochastic methods have been powerful in other areas as well, such as complicated financial and forecasting models that involve several random and difficult to assess input variables. There are different methods of stochastic analyses, some relying on analytical approaches for predicting outcomes, e.g., first and second-order reliability methods. Using statistical principles, these methods enable predicting outputs from randomly determined input variables, which often follow certain types of random patterns or probability distributions. Stochastic methods thus consider input data variation or fluctuation and enable precise outcome estimation, even for complex systems. It is important to recognize that in some applications, uncertainty in input data may be dependent on each other. Such problems would be very difficult to predict, however stochastic analysis enables simulating the outcome by performing a large number of simulations known as stochastic projections [33].

It is convenient to consider explicit input parameters when solving engineering problems. As such, explicit outputs are also simulated. However, strictly speaking, explicit input values do not exist for material properties, boundary conditions, and structural geometries, since such data may be subject to variation or fluctuation. Therefore, considering input parameters that are explicit can lead to generating hidden errors in final solutions and outcomes that are far from reality. Alternatively, by considering uncertainty for input parameters, robust results can be obtained. Moreover, the number of input parameters that influence output values increases as engineering models become more comprehensive. Consequently, material designers are faced with multiple challenges, which highlights the importance of developing innovative simulation methods.

Stochastic finite element analysis (SFEA) is a statistical method developed recently for solving sophisticated engineering problems that are impossible to solve using analytical methods [34,35]. As the name implies, SFEA is capable of considering uncertainty for model input parameters. In the present context of filler-modified polymers, several random variables need to be considered for material design, e.g., particle locations, orientations, dimensions, and properties. All of these parameters are known to affect outcomes appreciably. Given the complex nature of this problem, SFEA is considered a viable approach for material properties prediction.

Two different options are pursued connecting stochastic analysis with finite element analysis (FEA) in order to perform SFEA on engineering models. The first option entails the use of dedicated commercial FEA software with stochastic analysis and scripting capability [36,37,38]. Stochastic analysis and scripting capabilities are hereby integrated into the software, making it convenient for a user to perform SFEA since the software applies uncertainty to input variables. However, this first option is limited in terms of flexibility to incorporate specific modeling requirements. The second and arguably more powerful option involves advanced scripting for connecting stochastic analysis with FEA, providing maximum modeling flexibility, e.g., in terms of applying uncertainty to input variables, choosing a suitable pseudorandom number generator and performing customized FEA processes, examples of which are demonstrated by reference [39]. However, the second option is typically more challenging for users to perform. In addition, with greater opportunities for capturing complexities in terms of model features and input data variation, the second option is prone to become computationally expensive, necessitating the use of high-performance computational resources. Recently, the present authors developed an SFEA framework for predicting the effective thermal conductivity of particulate polymer composites [40]. The interested reader is referred to this article for detailed information about the developed SFEA framework, which is briefly summarized in the subsequent section. In general, this approach is capable and intended of modeling filler particle sizes ranging from nano- to micro-scales. However, especially for nano-scale fillers, great care must be taken to properly identify and implement the particle-to-particle and particle-to-matrix interactions in order to correctly capture the effective material properties. For the present study, the SFEA framework was employed to explore mechanical properties, i.e., elastic modulus, Poisson’s ratio, and coefficient of thermal expansion (CTE), for a composite with size-distributed spherical particles. Specifically, a composite consisting of an epoxy matrix with micro-sized spherical glass particles was studied and numerical results were contrasted with analytical predictions and experimental data from the technical literature.

## 2. Stochastic FEA Framework 

The developed modeling framework employs Monte Carlo simulation techniques and computes outcomes using statistical analysis, which necessitates large numbers of model iterations. A customized stochastic analysis process in conjunction with parametric FEA was thus developed that automated the process of applying uncertainties to input variables by connecting several program modules using multiple programming languages. The illustration in Figure 1 is the high-level architecture of the algorithm used for the modeling framework. In the following, the various modules depicted in Figure 1, including associated acronyms, will be highlighted.

Visual Basic for Applications (VBA) programming language was employed for connecting and routing information between the various modules used in the framework. A customized “Front End” form was created that enabled capturing of all input parameters required for the analysis and storing them in the database (‘DBMS’). Input parameters captured by the “Front End” form included: Size information for the modeled representative volume element (RVE), FVF that were to be analyzed, filler and matrix properties, filler particle size distribution, boundary conditions, settings controlling the contact behavior for filler particles and matrix, and parameters for finite element (FE) mesh generation.

An Open Database Connectivity (ODBC) method was used for developing the Database Management System (DBMS). The ODBC approach enabled creating an operating routine that was independent of the database and provides a high degree of flexibility for accessing the database at any time in the analysis. One task of the DBMS was the transfer of input parameters to the Monte Carlo Simulation (MCS) module in order to perform the SFEA.

Using VBA programming language, the MCS module was developed having a tabulated structure. The module enabled storing input parameters and results created by the model generation and analysis sub-processes. The algorithm as shown in Figure 2 is repeated for each specified FVF until statistical objectives are satisfied. The MCS module computes mean values, standard deviations, and variances, which were used as objective values for terminating the framework’s iteration. Once acceptance criteria were satisfied, e.g., an explicit standard deviation or variance, the MCS stops iterating and calculates the final effective mechanical properties. Results calculated in each iteration and mean values are saved to the database, which can be accessed through the “Front End”.

The Random Number Generator module (RNG, see Figure 1) was developed using the general mathematical programming environment MATLAB, enabling the generation of random numbers required for input variables. In the present study, these variables are Cartesian coordinates for particle locations (*X*, *Y*, *Z*) and the particle diameter. Particles were thus randomly distributed and dispersed within the RVE. The RNG module, illustrated in Figure 3, also performed collision detection between particles inserted into the RVE. A routine was defined that determined the distance for each particle newly inserted into the RVE with respect to all other particles already contained in the RVE. If no particle collision was detected, the new particle was accepted into the RVE; otherwise, the particle was rejected. This process continued until the specified FVF was satisfied, at which point the RNG module stops providing randomized particle data.

The composition of particulate filler materials may follow a certain particle size distribution. Simply using an average particle size may not reflect the behavior of the composite material. Hence, the SFEA framework was developed to account for particle size data that conforms to a prescribed size distribution using a data binning approach. The interested reader is referred to reference [40] for details on the algorithm that was included to conform particle dimensions to explicit size distribution, and for the influence on computational performance when adding size-ordered particles to the RVE. The latter was implemented in the SFEA framework.

For the model generation and analysis sub-processes, a customized parametric FEA platform was developed using the commercial FEA software ANSYS Workbench (Version 19, ANSYS Inc., Canonsburg, PA, USA). This FEA platform as depicted in Figure 4 was realized in conjunction with scripting in IronPython programming language. Two distinct submodules were developed in order to create the parametric three-dimensional model geometry and subsequently the full FEA model in a two-step process. In the first step, ANSYS DesignModeler in conjunction with scripting in JAVA programming language was used to read the data generated by the RNG module and create geometries for particles and the RVE. The resulting geometric representation was then transferred to the FEA modeling environment. For the second step, the FEA environment was developed using ANSYS Mechanical, again in combination with JAVA scripting, which facilitated an automated process of reading data from the database and creating the FEA model, including the application of material properties, mesh generation, and extraction of FEA results. Subsequent to mesh generation, the FEA environment submodule performed a convergence study by refining the mesh and extracting results to check whether results convergence was satisfactory. Finally, results obtained with appropriately refined meshing were transferred to the MCS module and saved for further statistical analysis.

## 3. Model Details

The developed SFEA framework was herein employed to predict the mechanical properties of particulate polymer composites with randomly distributed spherical particles under static-structural condition. Specifically, spherical glass bead particles embedded in epoxy were considered. Properties were adopted from the technical literature accordingly [41,42], see Table 1. Materials were treated as linear elastic. The particle size distribution illustrated in Figure 5 was adapted from reference [43] assuming that all particles fall within a band ranging from 0 to 50 μm and a data binning approach with 5 bins was used to have particle sizes emulate the given distribution. The size of the cubical RVE was set at 400 μm based on preliminary size effect studies.

Three-dimensional 10-node quadratic tetrahedral structural solid elements (SOLID187) were used for meshing both particles and matrix. Figure 6 illustrates an example mesh generated for the RVE and the particles included within it. The interface between the particles and matrix was defined by employing three-dimensional 8-node surface-to-surface contact elements (CONTA174 and TARGE170). In the present study, particle-matrix interfacial contact was assumed to constitute perfect bonding, and hence, no relative displacement at the interface was allowed.

Boundary conditions were applied to the RVE for generating strain and stress in the FE model. Nodes in one of two opposing RVE faces were restrained from out-of-plane displacement and rotation, while nodes remained free to move with the plane. This constraint was accomplished defining a cylindrical coordinate system at each respective node with the axial coordinate being aligned in the out-of-plane direction *ζ* and constraining the appropriate coordinate directions. Nodes in the opposing RVE face were subjected to a uniform displacement of 1 μm perpendicular opposite to the constrained RVE face. As such, a global normal strain *ε**_ζ_* of 2500 με was applied to the RVE with a size of 400 μm.

Using Equation (5) the modulus of elasticity was computed for each node *i* located in the RVE face subjected to the prescribed displacement.
(5)$$\sigma_{\zeta i}=E_{i}\varepsilon_{\zeta }$$
where *σ**_ζi_* and *E_i_* are the normal stress extracted from the FE results and modulus of elasticity on the *i*th node, respectively. As an example, Figure 7 displays normal stress on the model for a FVF of 0.45.

Invoking the Hencky strain method as given by Equation (6) for all nodes subjected to the prescribed displacement yields nodal strains being equivalent to the global strain.
(6)εζi=ln(liL)
where *l_i_* is the deformed RVE length at a specific node position and *L* the initial RVE length. 

Finally, the effective modulus of elasticity *E*_eff_ for the polymer composite embodied by the RVE was calculated using Equation (7).
(7)$$E_{\mathrm{eff}}=\frac{\sum_{i=1}^{n}E_{i}}{n}$$
where *n* is the total number of nodes.

In order to facilitate evaluating the effect of filler loading on Poisson’s ratio, boundary conditions were modified as compared to the modulus of elasticity analysis, that is, Cartesian boundary conditions were applied by restraining the out-of-plane displacement of nodes located in one of two opposing RVE faces, with nodes being free to move in the mutually perpendicular directions. To restrain rigid body motion of the RVE, one corner node of the restrained RVE face was held fixed in all three Cartesian directions. As in the previous case, nodes associated with the RVE face opposite to the constrained plane were again subjected to a 1 μm displacement normal to the plane.

Using Equation (8), Poisson’s ratio values *ν_i_* were calculated for each node in the displaced RVE face. The effective Poisson’s ratio *ν*_eff_ for the RVE was determined by Equation (9).
(8)$$\nu_{i}=\frac{\varepsilon_{\Psi i}}{\varepsilon_{\zeta i}}$$
(9)$$\nu_{\mathrm{eff}}=\frac{\sum_{i=1}^{n}\nu_{i}}{n}$$
where *ε**_Ψi_* and *ε**_ζi_* represent transverse and normal strain, respectively.

The effect of filler loading on the CTE was assessed by applying the same constraints to a single RVE face as for the modulus of elasticity analysis. In addition, a temperature change Δ*T* of 10 K was applied to the entire RVE. The CTE associated with node *i*, *α**_ζi_*, and the effective CTE for the RVE, *α*_eff_, were calculated using Equations (10) and (11), respectively. As an example, deformations normal to the RVE plane due to thermal expansion are depicted in Figure 8.
(10)$$\alpha_{\zeta i}=\frac{\Delta l_{\zeta i}}{L\;\Delta T}$$
(11)$$\alpha_{\mathrm{eff}}=\frac{\sum_{i=1}^{n}\alpha_{\zeta i}}{n}$$
where ∆*l**_ζi_* is axial displacement extracted from the FE results at the *i*th node.

## 4. Results and Discussion

Prior to studying the effect of filler addition on mechanical properties, test cases were performed to assess the distribution of filler particles within RVE. The particle coordinates for the three Cartesian directions were plotted as depicted in the example shown in Figure 9. It can be observed that particles were randomly distributed and dispersed within RVE. Figure 9 also illustrates that, as mentioned above, the largest particles were added to the RVE first (smallest particle index), followed by particles in decreasing size order. Moreover, three sets of numerical analyses were performed, and mechanical properties were calculated for each of the mutually perpendicular axes defining the RVE (*X*, *Y*, *Z*). The example data in Table 2 indicates that differences between the results for given directions are negligible. Hence, it was concluded that the SFEA framework, and especially the RNG module’s random number generator, performed satisfactorily for creating true randomness in the analysis.

After confirming its proper function, the SFEA framework was employed to predict the effective modulus of elasticity, Poisson’s ratio, and CTE for the glass bead/epoxy composite for FVF of 0.05, 0.10, 0.15, 0.20, 0.275, 0.35, and 0.45. As mentioned above, the effective properties for each volume fraction were computed via the MCS module and then transferred to the database for statistical analysis. The unbiased standard deviation, variance, and mean were calculated for the stored effective properties data. The MCS module iterates the FEA model until the acceptance criteria defined in the MCS module were satisfied, i.e., the standard deviation reaches an explicit value. Then, the final effective properties for a given FVF were computed as the average from all iterations.

After completing a full Monte Carlo simulation for a specific FVF, a study was performed to assess the quality of the data that was acquired. It was hypothesized that the SFEA framework creates true randomness with data falling under a normal distribution with close congruence between the mean and median values. Ideally, the determination of properties for each FVF would be done with an infinite number of iterations, and hence, predictions were considered continuous random variables for statistical analysis purposes. A probability density function (PDF) was thus computed for each dataset related to a given FVF. The normalized PDF for modulus of elasticity, Poisson’s ratio, and CTE data are depicted in Figure 10a, Figure 11a and Figure 12a.

The data presented in Figure 10, Figure 11 and Figure 12 suggests that the effective properties are normally distributed. To test this assumption, statistical data analyses were performed based on mean, median, skewness, and kurtosis values, with skewness and kurtosis providing a measure for asymmetry and peakedness of the distributions, respectively. In other words, these measures were employed to assess how well the data adhere to a normal distribution (e.g., skewness and kurtosis of zero indicate a perfect normal distribution). A comprehensive study was performed by West et al. [46] regarding normal distribution quality indicators. Based on their study, skewness greater than two and kurtosis greater than seven were considered indicators for a significant deviation from normal distribution. In addition, the number of iterations was emphasized as a parameter that had the greatest influence on skewness and kurtosis. Table 3, Table 4 and Table 5 are showing correspondingly the statistical analyses performed for modeling data relating to the modulus of elasticity, Poisson’s ratio, and CTE. The statistical analysis results show that mean and median values are closely congruent, and skewness and kurtosis values are close to zero. Therefore, all datasets were considered to have passed normality test requirements, which is also a confirmation that sufficient numbers of iterations were performed.

As mentioned above, calculated final effective properties can be considered continuous random variables, and hence, the probability of occurrence of a specific value can be calculated within an interval as expressed by Equation (12). Data from the model simulations can thus be presented in the form of cumulative distribution functions (CDFs). CDFs for the effective properties, i.e., the modulus of elasticity, Poisson’s ratio, and CTE, are shown in Figure 10b, Figure 11b and Figure 12b, respectively.
(12)$$P\left(a\leq Χ\;\leq b\right)=\;\int_{a}^{b}f\left(\chi \right)d\chi$$
where *P* is the probability of an effective property occurring within an interval *a* and *b*; *Χ* and *f*(*χ*) are a continuous random variable and the PDF, respectively. 

In a final step of the modeling data analysis, results for the effective modulus of elasticity and Poisson’s ratio were compared with experimental results published in the technical literature [41,42], as shown in Figure 13 and Figure 14, respectively. For the modulus of elasticity, the experiments indicated a non-linear increase with increasing FVF. It can be observed that the numerical data are in good agreement with the experiments. In fact, the predictions were within the experimental data scatter. Figure 13 also includes predictions from analytical models, namely, the Mori-Tanaka and Hashin-Shtrikman approaches. It can be seen that both analytical methods delivered similar results yet underpredict the values from experiments and numerical modeling, especially for higher FVF.

Referring to Figure 14, experimental data indicated a reduction in Poisson’s ratio with rising FVF. The numerical predictions followed the experimental values rather well up to an FVF of 0.15, at which point a shift in the experimental data seemed to have occurred compared to its initial trend. The original publication [42] from which the experimental data were sourced from did not address nor further investigate this behavior. It is herein speculated that at higher FVF, the sample morphology may have deviated from the assumptions made in the present study, which is a random filler distribution and dispersion, leading to a reduced ability to lower the Poisson’s ratio.

Numerical predictions for effective CTE are depicted in Figure 15. No relevant experimental data could be found for comparison in the technical literature. Alternatively, a basic Voigt model prediction was made as shown in the figure for comparison with the numerical modeling results. Both the numerical and analytical predictions describe a significant reduction of CTE with increasing FVF. It can be observed that the graph associated with the numerical model was slightly non-linear, predicting a 6% lower CTE for the highest filler loading compared to the Voigt model. Interestingly, the rather basic Voigt model appeared to yield acceptable predictions in this context. However, prior to generalizing such a statement, it would be prudent to study particle size effects and compare data with experimental results.

Similar to other research work, the present study confirms that filler particle and polymer matrix properties, as well as filler loading, significantly affect the effective composite properties. It has been shown that particle size is another important parameter since polymer composites created with smaller particles tend to increase the effective modulus of elasticity to a greater extent compared to composites containing larger particles at the same FVF [47]. Most analytical approaches are unable to capture the influence of filler geometry and, therefore, fail to predict properties accurately [41,48]. Alternatively, empirical models can be fitted to predict the properties of specific composite configurations [49]. While such an approach may increase the accuracy for material property predictions, it has drawbacks in terms of requiring a range of empirical models with limited applicability. The SFEA framework that was applied in the present study alleviates many of the shortcomings associated with analytical and empirical techniques. The SFEA framework, while being computationaly more intensive than analytical and empirical methods, was shown to be versatile in terms of filler and matrix material characteristics and geometries. Moreover, the developed numerical technique is capable of uniting the prediction of mechanical as well as other physical properties [40] in one model, further adding to the versatility of the presented approach. The SFEA framework is thus seen as an attractive tool for material designers. Its application in conjunction with experimental validation may thus serve to accelerate material development to meet application requirements. 

## 5. Conclusions

A stochastic finite element analysis framework was developed to enable the property prediction of particulate filler modified polymer composites. The employed modeling approach is based on Monte Carlo principles and allows for capturing the effects of filler size distribution, filler shapes, and orientations, which are important parameters for accurately predicting composite properties. While the developed analysis framework is capable of predicting a variety of mechanical and physical properties, such as thermal conductivity, the present study focused on predicting the effective modulus of elasticity, Poisson’s ratio, and coefficient of thermal expansion for the case of randomly distributed and dispersed spherical glass particles embedded in an epoxy polymer matrix. 

The numerical results obtained from the analysis framework were compared with experimental values and analytical approaches. The numerical data were found to agree well with non-linear trends observed in the experiments, especially for the elastic modulus, in which case predictions fell within the experimental data scatter. Numerical predictions deviated from experimental Poisson’s ratio data for filler volume fractions exceeding 0.15. This may be a result of the morphology changes in test specimens at higher filler loadings, e.g., an increasing extent of filler agglomerations, which would deviate from assumptions made for the numerical model. Nevertheless, in general, the data from the numerical model were found to predict the experiments rather well, while the employed analytical methods were less successful in predicting the test data accurately.

This study shows that numerical techniques, including the current stochastic finite element analysis framework, are attractive and effective for modeling particulate filler modified polymer composites, outperforming analytical methods in terms of versatility and purely experimental campaigns with regard to time and cost. Given that the developed analysis framework is capable of treating binary, ternary, and higher order polymer composites with randomly distributed filler particles, it is seen as a suitable tool for material designers to achieve greater accuracy predictions and use less costly and time-consuming experimentation in order to expedite the development of advanced filler modified polymer composites.

## Figures and Tables

**Figure 1 materials-12-02777-f001:**
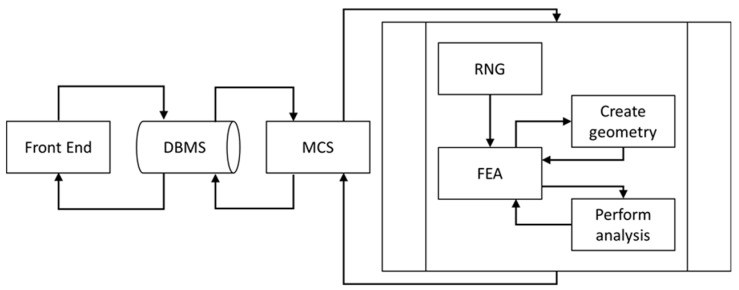
High-level architecture of developed stochastic finite element analysis (FEA) framework [40].

**Figure 2 materials-12-02777-f002:**
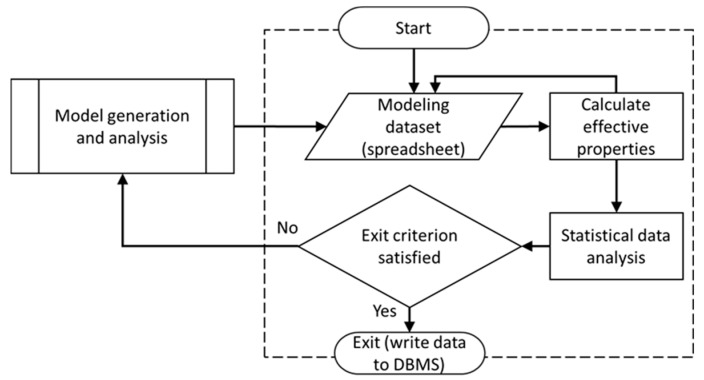
Schematic representing the algorithm of Monte Carlo Simulation (MCS) module [40].

**Figure 3 materials-12-02777-f003:**
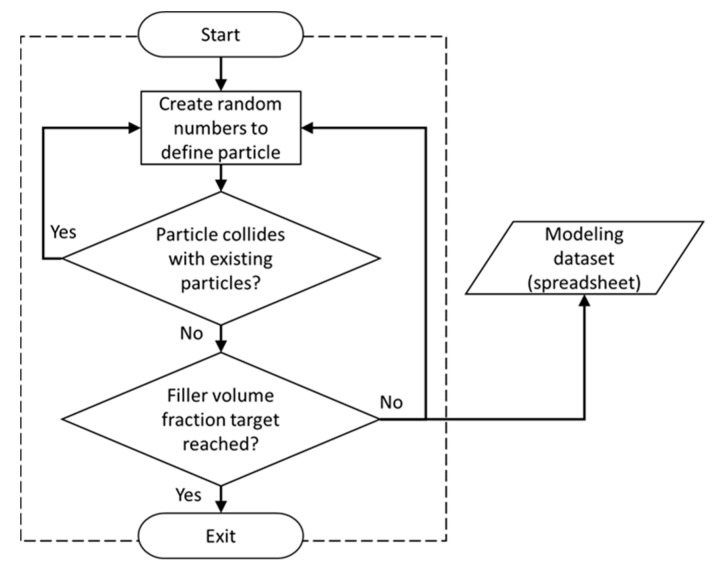
Schematic illustrating the Random Number Generator (RNG) module [40].

**Figure 4 materials-12-02777-f004:**
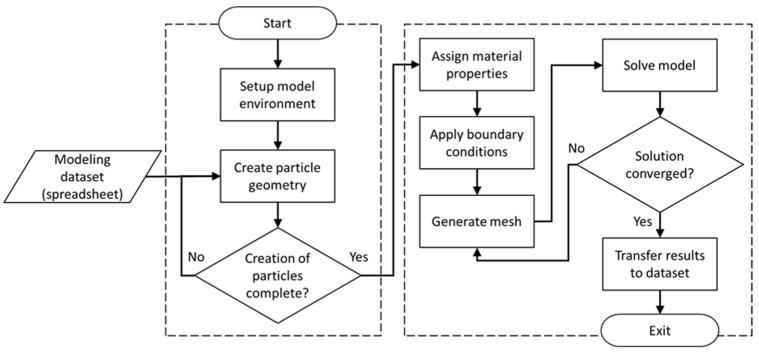
Schematic of the FEA platform for model generation and analysis sub-processes [40].

**Figure 5 materials-12-02777-f005:**
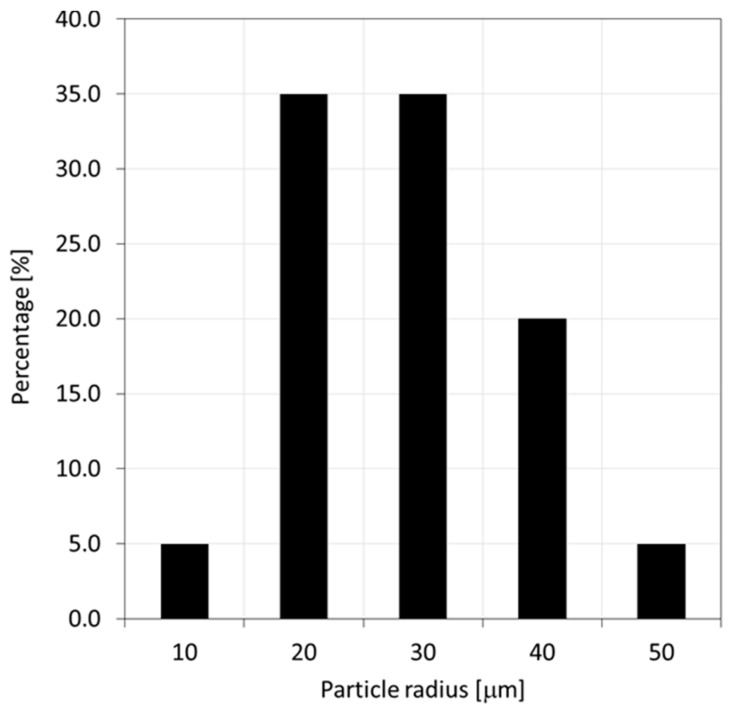
Glass bead size distribution adapted from reference [43].

**Figure 6 materials-12-02777-f006:**
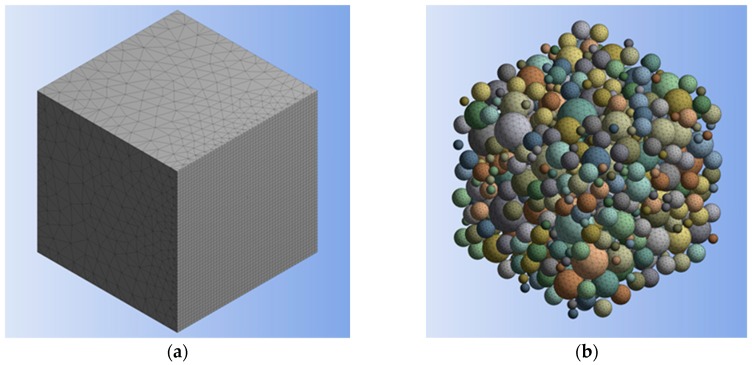
Example meshing of (**a**) representative volume element (RVE) and (**b**) particles.

**Figure 7 materials-12-02777-f007:**
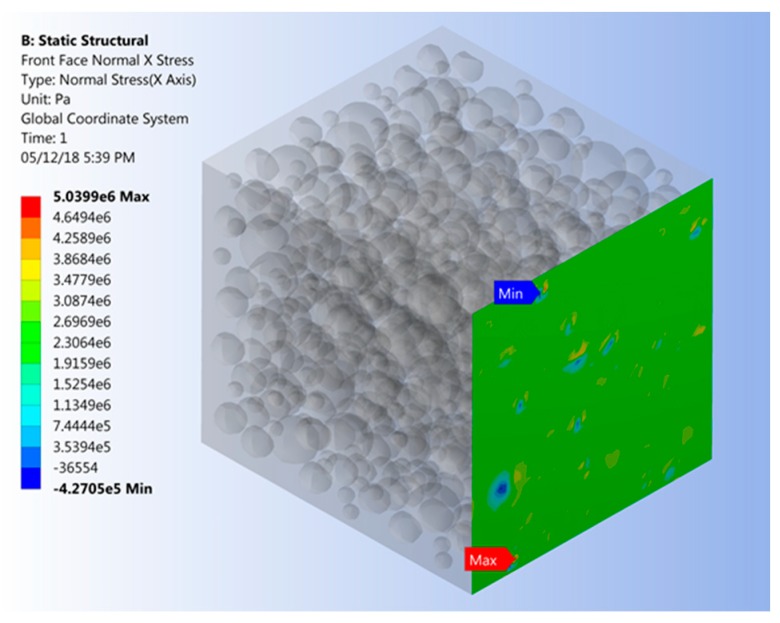
Finite element model depicting normal stress on RVE face being subjected to uniform nodal displacement.

**Figure 8 materials-12-02777-f008:**
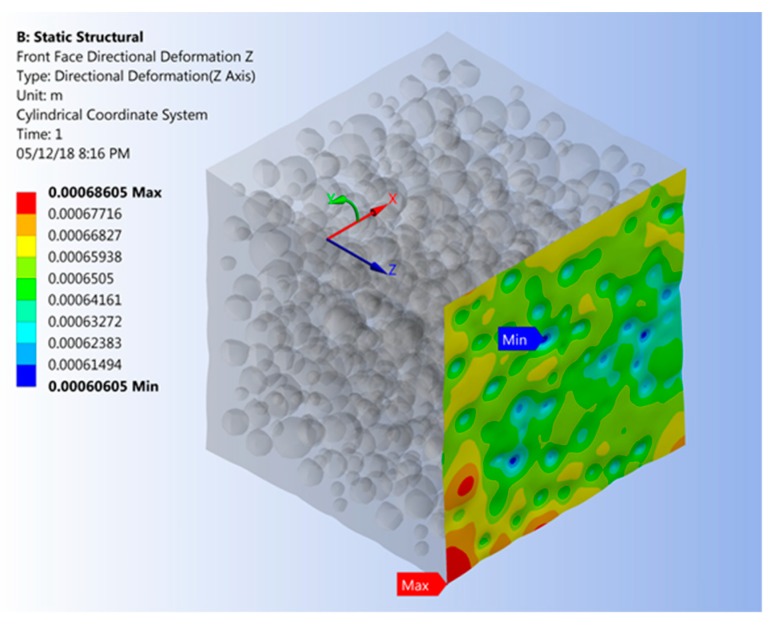
Deformations due to thermal expansion normal to the RVE plane (deformations scaled).

**Figure 9 materials-12-02777-f009:**
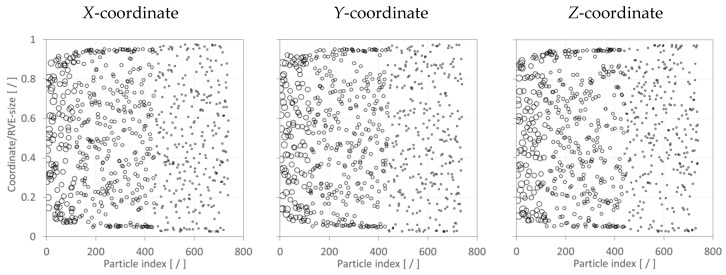
Distribution of particles in RVE for 0.45 FVF. Symbol size is indicative of particle size.

**Figure 10 materials-12-02777-f010:**
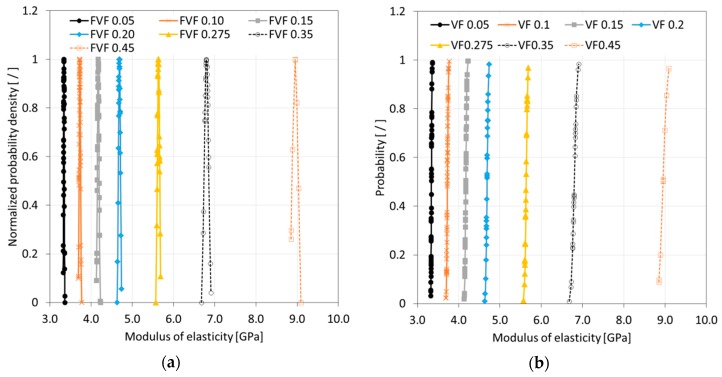
Normalized probability density (**a**) and cumulative distribution function (**b**) of modulus of elasticity data for simulated glass bead/epoxy composites.

**Figure 11 materials-12-02777-f011:**
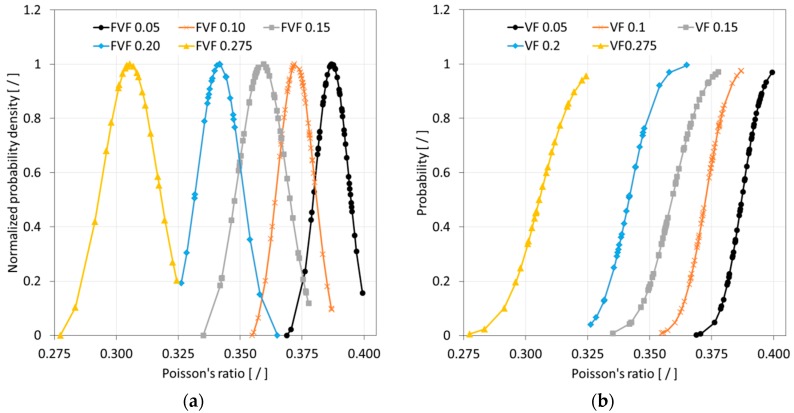
Normalized probability density (**a**) and cumulative distribution function (**b**) of Poisson’s ratio data for simulated glass bead/epoxy composites.

**Figure 12 materials-12-02777-f012:**
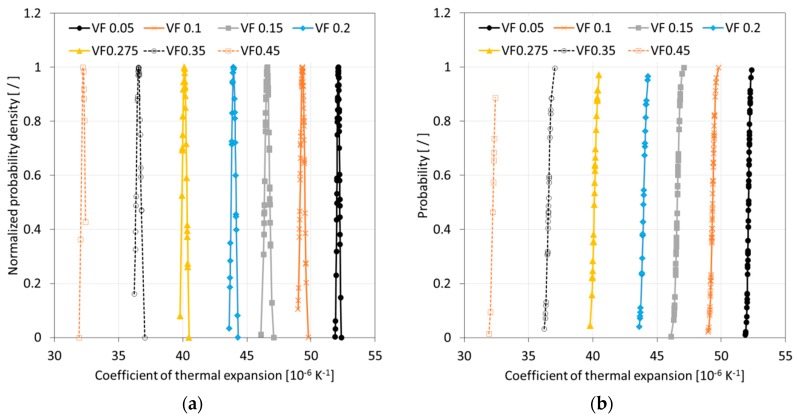
Normalized probability density (**a**) and cumulative distribution function (**b**) of the coefficient of thermal expansion data for simulated glass bead/epoxy composites.

**Figure 13 materials-12-02777-f013:**
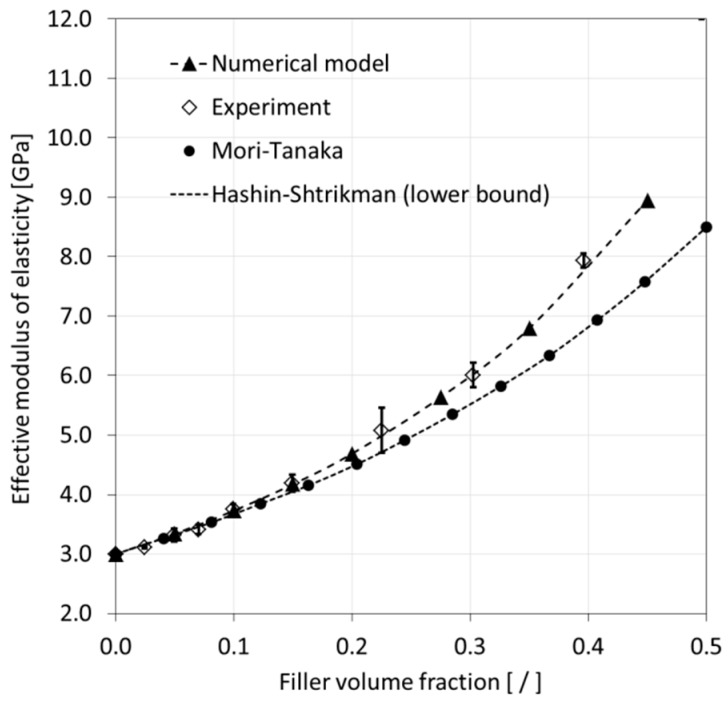
Comparison of modeling results for the modulus of elasticity with analytical model predictions and experimental data [41,42]. Error bars express experimental data scatter.

**Figure 14 materials-12-02777-f014:**
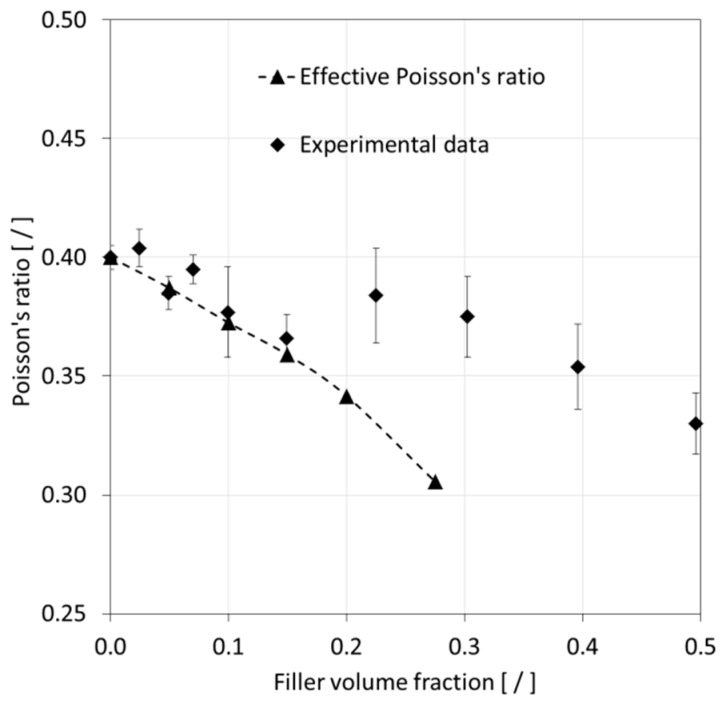
Comparison of modeling results for the Poisson’s ratio with experimental data [42]. Error bars express experimental data scatter.

**Figure 15 materials-12-02777-f015:**
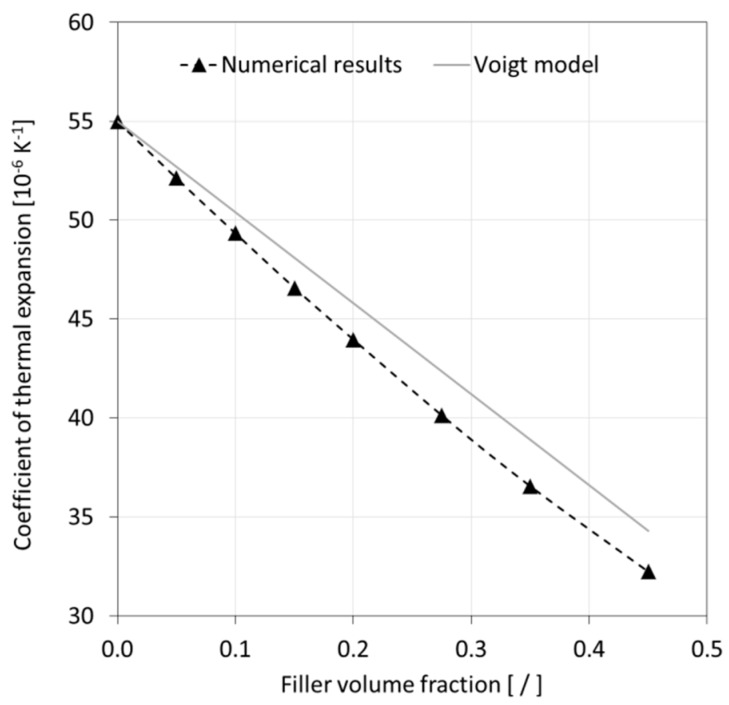
Comparison of modeling results for the coefficient of thermal expansion with Voigt model predictions.

**Table 1 materials-12-02777-t001:** Properties of filler particles and polymer matrix.

Property	Glass Beads	Epoxy Matrix
Modulus of elasticity (GPa) [41,42]	76.0	3.0
Poisson’s ratio (/) [41,42]	0.23	0.40
Coefficient of thermal expansion (K^−1^) [44,45]	9.0 × 10^−6^	55.0 × 10^−6^

**Table 2 materials-12-02777-t002:** Properties calculated along mutually perpendicular axes defining the RVE for 0.275 FVF.

Property	*X* Direction	*Y* Direction	*Z* Direction
Modulus of elasticity (GPa)	5.642	5.639	5.642
Poisson’s ratio (/)	0.3055	0.3035	0.3170
Coefficient of thermal expansion (10^−6^ K^−1^)	40.13	39.94	39.91

**Table 3 materials-12-02777-t003:** Statistical analysis of modeling results for the modulus of elasticity.

**FVF**	0.05	0.10	0.15	0.20	0.275	0.35	0.45
**Number of Simulations (/)**	50	50	50	25	25	25	10
**Mean Value (GPa)**	3.340	3.728	4.173	4.688	5.635	6.798	8.951
**Median Value (GPa)**	3.340	3.728	4.173	4.690	5.633	6.791	8.952
**Standard Deviation (GPa)**	0.010	0.017	0.020	0.024	0.028	0.053	0.075
**Skewness (/)**	0.381	0.186	0.361	−0.288	−0.179	−0.065	0.302
**Kurtosis (/)**	−0.234	−0.017	0.807	0.516	−0.452	0.570	−0.157

**Table 4 materials-12-02777-t004:** Statistical analysis of modeling results for the Poisson’s ratio.

**FVF**	0.05	0.10	0.15	0.20	0.275
**Number of simulations (/)**	50	50	50	25	25
**Mean Value (GPa)**	0.3871	0.3725	0.3589	0.3416	0.3055
**Median Value (GPa)**	0.3872	0.3742	0.3568	0.3406	0.3044
**Standard Deviation (GPa)**	0.0065	0.0073	0.0098	0.0087	0.0110
**Skewness (/)**	−0.561	−0.394	−0.001	0.762	−0.617
**Kurtosis (/)**	0.387	0.149	−0.347	1.169	0.865

**Table 5 materials-12-02777-t005:** Statistical analysis of modeling results for the coefficient of thermal expansion.

**FVF**	0.05	0.10	0.15	0.20	0.275	0.35	0.45
**Number of Simulations (/)**	50	50	50	25	25	25	10
**Mean Value (10^−6^ K^−1^)**	52.12	49.34	46.58	43.96	40.17	36.55	32.24
**Median Value (10^−6^ K^−1^)**	52.14	49.33	46.59	43.96	40.19	36.54	32.28
**Standard Deviation (10^−6^ K^−1^)**	0.015	0.024	0.026	0.038	0.034	0.035	0.046
**Skewness (/)**	−0.283	0.164	0.179	−0.010	−0.114	0.479	−1.410
**Kurtosis (/)**	−0.044	0.170	0.665	−0.822	−0.490	1.00	1.681

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
