# Peer review of "Stochastic Finite Element Analysis Framework for Modelling Mechanical Properties of Particulate Modified Polymer Composites"

_materials, 2019, doi:10.3390/ma12172777_

Round 1

Reviewer 1 Report

 Title of paper: Stochastic Finite Element Analysis Framework for Modelling Mechanical Properties of Particulate Modified Polymer Composites

This is an interesting work which can be considered for publication in Materials after minor revision as follows:

1.    Some quantitative statements have to be provided in the abstract and conclusion.

2.    Quality of the discussion section has to be improved.

3.    There are few grammatical errors in the manuscript which need to be corrected.

Author Response

Dear Reviewer,

Thank you for your constructive feedback stipulating minor revisions, which we carefully considered and implemented in order to improve our manuscript. Specifically, we changed the document in the following manner.

1. "Some quantitative statements have to be provided in the abstract and conclusion."

The abstract and conclusions were expanded to include information on the congruence between modeling and experimental results.

2. "Quality of the discussion section has to be improved."

The discussion section was critically reviewed and various changes were made to improve clarity.

3. "There are few grammatical errors in the manuscript which need to be corrected."

The full document was carefully reviewed, and several grammatical errors were indeed found. We apologize for these oversights in the original manuscript. The errors were remedied.

Note that the changes in the manuscript (besides minor editorial improvements in terms of grammar etc.) are highlighted by red text in the revised document.

Best regards,

PM

Reviewer 2 Report

Authors present results from application of a stochastic finite element approach to model polymer-based nanocomposites. While the topic of polymer nanocomposites is of paramount importance for materials and polymer science the current manuscript appears to be too specialized to be published in Materials.

There exist some important flaws: for example the abstract is too general focuses on standard knowledge and provides very little insight of the current contribution with respect to established know-how.

Similarly, the paper discusses for too long information that is basic information like for example the algorithm to generate non-overlapping configurations of particles of varied size inside the polymer matrix. Same stands for the schemes of the algorithmic flow, which appear to be very basic (RNG module etc).

Author Response

Dear Reviewer,

Thank you for your feedback, which we considered carefully. We would like to respectfully provide the following rebuttal to your review comments.

1. “...the current manuscript appears to be too specialized to be published in Materials.”

According to the ‘Aims & Scope’ of Materials, theoretical works, i.e., modeling, are welcome as contributions. Specifically, ‘materials engineering’ is listed as topics covered by Materials. We consider material design, which is purpose of the current modeling approach, to be part of materials engineering. Moreover, composite materials are one of the topics stated in the journal’s ‘Aims & Scope’. Based on this information, we feel that the paper is well suited for Materials.

2. “...the abstract is too general focuses on standard knowledge and provides very little insight of the current contribution with respect to established know-how.”

Based on our study of the technical literature, we feel that improved and more comprehensive modeling techniques are needed. The abstract points out the benefits of our developed modeling approach, i.e., “employing stochastic finite element analysis, reduces efforts for assessing material properties compared to experimental work while increasing the level of accuracy compare to other available approaches, such as analytical methods.” We would like to draw attention on the term ‘stochastic’. We feel that the stochastic nature of the modeling approach is not only original but also a powerful method. Please note that we modified the abstract to highlight the findings related to the comparison of modeling results to experimental data.

3. “...the paper discusses for too long information that is basic information like for example the algorithm to generate non-overlapping configurations of particles of varied size inside the polymer matrix. Same stands for the schemes of the algorithmic flow, which appear to be very basic (RNG module etc).”

We respectfully disagree with the notion that the presentation of the modeling algorithm is basic and thus superfluous. In order for a reader to sufficiently appreciate and possibly adopt or reproduce the modeling results, essential modeling steps need to be presented in the given detail. As alluded to in the manuscript, the model entails many intricate processes (e.g. the addition of size-distributed particles), which have been substantially abbreviated to order to keep the manuscript as succinct as possible.

Further note that changes were made to the manuscript based in all reviewers’ comments. These changes are highlighted by red text in the revised document (besides minor editorial improvements in terms of grammar etc.).

Best regards,

PM

Reviewer 3 Report

In this paper, mechanical properties (elastic modulus, Poisson’s ratios, and thermal expansion coefficient) of polymeric composites were analyzed using a numeric modeling framework. The authors previously reported the effective thermal element conductivity of polymer composites by the stochastic finite element analysis (SFEA) framework as ref. 40. In this paper, the method of SFEA framework was extended to the above physicochemical properties of polymeric composites. As the polymer composites, polymeric materials with size distributed spherical particles were investigated. Moreover, the calculated parameters were compared with that of the experimental results of other reference data. However, the discussion of the comparison between SFEA framework and experimental results was not enough. Therefore, the manuscript will be suitable for this journal after minor revision. My questions and comments are shown as follow.

Q1. Can the SFEA framework use the nano-particles as the distributed particles in the composite materials? The nano-particles having the huge surface area might possess different properties as compared with that of micro-particles.

Q2. Can you predict new properties of the composite materials via the SFEA framework?

Q3. How did you apply the parameters of polymer such as molecular weight, molecular weight distributions, and the degrees of crystallization to the SFEA framework?

Q4. You should compare the calculated parameters with more data of experimental results.

Author Response

Dear Reviewer,

Thank you for your constructive feedback stipulating minor revisions, which we carefully considered and implemented in order to improve our manuscript. Specifically, we changed the document in the following manner.

1. The full document was carefully reviewed, and several grammatical errors were remedied. We apologize for these oversights in the original manuscript.

2. "However, the discussion of the comparison between SFEA framework and experimental results was not enough."

The discussion section was critically reviewed and various changes were made to improve clarity.

3. "Q1. Can the SFEA framework use the nano-particles as the distributed particles in the composite materials? The nano-particles having the huge surface area might possess different properties as compared with that of micro-particles."

In general, yes, the developed SFEA framework is capable of modeling not only micro-size fillers but also those on the nano-scale. However, especially for nano-scale fillers, great care must be taken to properly define the particle-to-particle and particle-to-matrix interactions, in order to properly capture the physics of the material system. A statement clarifying this query has been added to the Introduction section of the manuscript. It may be of interest to the reviewer that the authors are in the process of concluding a study on the electrical properties of nano-composites utilizing the developed SFEA framework. However, this latest work is beyond the scope of the present manuscript.

4. "Q2. Can you predict new properties of the composite materials via the SFEA framework?"

This is a very interesting comment by the present reviewer, which we discussed extensively. In general, it is conceivable that new material properties can be predicted once the underlying physical conditions are known to an extent that they can be implemented in the FEA model. However, we feel that including such a claim in the present manuscript is too speculative, and further research on this subject much be completed first.

5. "Q3. How did you apply the parameters of polymer such as molecular weight, molecular weight distributions, and the degrees of crystallization to the SFEA framework?"

The SFEA framework is not a molecular dynamics simulation, and as such, the aforementioned parameters are not explicitly required. The matrix is considered a continuum. Nevertheless, once again, this reviewer raised a very valuable point. Including these parameters in some manner may provide for greater fidelity in future studies, such as modeling thermoset cure kinetics and residual properties (such as residual stress). However, we consider this an significant extension of the current modeling approach, and hence, inclusion of such notion is again too speculative for the current manuscript. 

6. "Q4. You should compare the calculated parameters with more data of experimental results."

While additional comparisons are possible (in addition to the presented two sources from the technical literature), such extension would significantly lengthen the manuscript while only providing minor additional information. The authors feels that they showed sufficiently that the SFEA framework is capable of predicting effective material properties as compared to well-accepted analytical models and some experimental works. In the interest of brevity, further comparisons were not made, especially since these typically require further extensive discussion (as shown in the case of the Poisson's ratio data in the present manuscript where property predictions deviated from experimental measurements for high filler loadings).

Note that the changes in the manuscript (besides minor editorial improvements in terms of grammar etc.) are highlighted by red text in the revised document.

Best regards,

PM